# Booster Dose of SARS-CoV-2 mRNA Vaccine in Kidney Transplanted Patients Induces Wuhan-Hu-1 Specific Neutralizing Antibodies and T Cell Activation but Lower Response against Omicron Variant

**DOI:** 10.3390/v15051132

**Published:** 2023-05-09

**Authors:** Andrea Del Mastro, Stefania Picascia, Luciana D’Apice, Maria Trovato, Pasquale Barba, Immacolata Di Biase, Sebastiano Di Biase, Marco Laccetti, Antonello Belli, Gerardino Amato, Potito Di Muro, Olga Credendino, Alessandra Picardi, Piergiuseppe De Berardinis, Giovanna Del Pozzo, Carmen Gianfrani

**Affiliations:** 1AORN A. Cardarelli–Internal Medicine Division 1–Immunology Unit, 80131 Naples, Italy; 2Institute of Biochemistry and Cell Biology, Italian National Council of Research, 80131 Naples, Italy; 3MeriGen Diagnostic&C, SAS, 80131 Naples, Italy; 4AORN A. Cardarelli–Clinical Pathology Division, 80131 Naples, Italy; 5AORN A. Cardarelli–Nephrology and Dialysis Unit, 80131 Naples, Italy; 6AORN A. Cardarelli–Molecular Biology Laboratory–Hematology and HSC Transplantation Unit, 80131 Naples, Italy; 7Institute of Genetics and Biophysics, Italian National Council of Research, 80131 Naples, Italy

**Keywords:** transplantation, kidney transplanted patients, COVID-19, SARS-CoV-2, Omicron variant, mRNA vaccine, neutralizing antibodies, T cells, immunity

## Abstract

Kidney transplanted recipients (KTR) are at high risk of severe SARS-CoV-2 infection due to immunosuppressive therapy. Although several studies reported antibody production in KTR after vaccination, data related to immunity to the Omicron (B.1.1.529) variant are sparse. Herein, we analyzed anti-SARS-CoV-2 immune response in seven KTR and eight healthy controls after the second and third dose of the mRNA vaccine (BNT162b2). A significant increase in neutralizing antibody (nAb) titers were detected against pseudoviruses expressing the Wuhan-Hu-1 spike (S) protein after the third dose in both groups, although nAbs in KTR were lower than controls. nAbs against pseudoviruses expressing the Omicron S protein were low in both groups, with no increase after the 3rd dose in KTR. Reactivity of CD4^+^ T cells after boosting was observed when cells were challenged with Wuhan-Hu-1 S peptides, while Omicron S peptides were less effective in both groups. IFN-γ production was detected in KTR in response to ancestral S peptides, confirming antigen-specific T cell activation. Our study demonstrates that the 3rd mRNA dose induces T cell response against Wuhan-Hu-1 spike peptides in KTR, and an increment in the humoral immunity. Instead, humoral and cellular immunity to Omicron variant immunogenic peptides were low in both KTR and healthy vaccinated subjects.

## 1. Introduction

Immune memory to SARS-CoV-2 is induced by booster vaccinations with either BNT162b2 or mRNA-1273 mRNA vaccines (from Pfizer/Biontech and Moderna, respectively), providing protection from severe coronavirus disease 2019 (COVID-19) by harnessing both humoral and cellular immune responses [1].

During the COVID-19 pandemic, many variants of SARS-CoV-2, from Alpha to Omicron, have arisen. These variants carry several mutations in the spike (S) protein that facilitate escape from immunity conferred by vaccines expressing and/or delivering the S protein from the ancestral SARS-CoV-2 Wuhan-Hu-1 strain [2]. One of the most recent variants, the B.1.1.529 (Omicron), is associated with a less severe disease although with a higher infectivity [3]. In vaccinated people, neutralizing antibody (nAb) titers against Omicron are lower compared to those against Wuhan-Hu-1 strain, most likely due to the occurrence of 15 mutations in the S receptor-binding domain (RBD) [4]. However, the majority of T cell epitopes are conserved in Omicron variant, and current vaccines provide a T cell mediated immunity against this largely spread variant. Specifically, 84–85% of CD4^+^ and CD8^+^ T cell responses are preserved against Omicron antigens over six months after the second dose of mRNA vaccine [5,6].

Other important factors affecting SARS-CoV-2 protective immunity are: (i) age (in older individuals the immune response is altered) [5], and (ii) presence of morbidities, such as onco-hematologic diseases, metabolic dysfunctions, autoimmunity, or immunodepression [7,8,9]. In addition, preexisting cross-reactive T cell immunity caused by a prior exposure to common cold coronaviruses sharing significant homology with SARS-CoV-2 may provide protection against infection and/or severe COVID-19 clinical course [10].

The immune system of kidney transplanted recipients (KTR) is impaired due to prolonged immunosuppressive treatment that aims at preventing graft rejection, increasing susceptibility to severe SARS-CoV-2 infection and mortality compared to general population [11]. Although it has been reported that the SARS-CoV-2 mRNA vaccine confers seropositivity in most KTR, immune response after two doses of vaccine results impaired in these immunocompromised patients and the third dose represents an essential requirement to provide immune protection [12]. The booster shot induced production of nAbs against the ancestral SARS-CoV-2 strain also in primary non-responder KTR subjects [13,14,15], although some of the recipients did not mount a detectable humoral immune response following the 3rd dose [16], while nAb titers were lower against Alpha, Beta, Delta and Omicron variants in both KTR and the general population [17,18], as confirmed by pseudotyped lentivirus [19] and live virus-based tests [20]. Moreover, the role of T cells in conferring protection has been underlined [21,22].

Recently, it has been demonstrated that the low neutralizing anti-RBD IgG response is dependent on the T-B cell interactions that took place in germinal centers, and it is influenced by immunosuppressive drugs [23]. These findings are supported by the analysis of immune responses to SARS-CoV-2 mRNA vaccines in lymph nodes of KTR. An impairment of germinal center function caused a decrease in RBD-specific memory B cells and nAb synthesis, probably as a consequence of lack of an adequate follicular CD4^+^ T cell help [24].

Several studies investigated the cellular immune response in KTR assessing spike-specific IFN-γ production by ELISPOT assay [21,25] or ELISA quantization [26,27]. Here, we evaluated humoral and cellular immune responses in a small cohort of KTR and healthy individuals after the 2nd and 3rd dose of BNT162b2 Pfizer-BioNTech vaccine. In detail, we tested serum samples from vaccinated individuals against pseudotyped lentiviruses bearing the S protein from the ancestral Wuhan-Hu-1 strain or the Omicron variant. We also carried out functional T cell assay [28] after stimulation with peptide pools spanning the S protein from either Wuhan-Hu-1 strain or Omicron variant, in order to reveal the lymphocyte activation status, and to further correlate the magnitude of T and B cell responses.

## 2. Materials and Methods

### 2.1. Cohort Description and Sample Collection

The EVADI-COVID-19 project is a single center cohort study which enrolled 106 adult subjects (43 healthy, 63 fragile patients) undergoing SARS-CoV-2 vaccination program with mRNA-based SARS-CoV-2 (Pfizer-BioNTech) vaccine at Cardarelli Hospital in Naples, from March 2021 to March 2022 (Figure 1A). Patients with a previous history or signs of COVID-19, or resulting positive for SARS-CoV-2 during vaccine administration, were excluded. Some of the patients did not receive all three doses of vaccines as they showed symptoms related to their pathology. For some subjects the amount of blood sample was not sufficient to perform both humoral and T cell assays.

The study was approved by the Cardarelli Hospital Ethical Committee (protocol nr 05/21 signed on 31 March 2021) and by the National Italian Spallanzani Ethical Committee (protocol nr 55, register 2022), and aimed to investigate the immune response to SARS-CoV-2 over time after vaccination in KTR patients with impaired immune system.

Fragile patients were defined according to the official description adopted by the Italian Ministry of Health and International Health Institutions for the organization of vaccine programs, indicating people aged ≥70 years and/or with chronic diseases, e.g., diabetes, hypertension, chronic obstructive pulmonary disease, and/or onco-hematologic diseases, and/or immunodepression. Multiple sclerosis patients treated with fingolimod or beta-interferon, Hodgkin, and non-Hodgkin lymphoma treated with anti-CD20 biologic drugs were excluded from the enrollment program.

This study focused on kidney transplanted patients (7 KTR: 4 F, 3 M, average age 44 ± 19 years, range 20–73, median: 48, interquartile range IQR: 57.00–30.00) and healthy donors (8 H: 4 F, 4 M, average age 48 ± 12 years, range 33–68, median: 46, IQR: 60.50–38.00), recruited among general population participating in the Cardarelli Hospital vaccination campaign (Table 1). The underlying diseases leading to transplant in KTR group were: arterial hypertension (3 patients, 2 females, 1 male), immune nephritis (2 patients, 1 female, 1 male; IgA-related in 1 patient), Autosomic Dominant Polycystic Kidney Disease (ADPKD, 1 patient, female), and obstructive renal failure (1 patient, male). All of the KTR patients regularly undergo periodic visits at our Nephrology Division with complete clinical evaluation, i.e., physical examination with measurement of vital parameters, peripheral blood, and urine sampling with measurement of multi-organ function parameters including serum creatinine, 24-h urine creatinine, serum concentration of immunosuppressors, update of medical history, modifications of treatments, instrumental exams for monitoring of infectious, cardiovascular and oncologic complications. No patient was affected by diabetes, thus we could exclude an additional factor of such complications. According to the current guidelines [29], in order to monitor post-transplant kidney function, creatinine clearance on 24-h urine samples is measured at each visit, with calculation of glomerular filtration rate (GFR). Based on our data, 100% of KTR had GFR normal (G1-T) or mildly-moderately decreased (G2-T, G3a-T). Immunosuppressors blood concentration was in normal range at each time point. Healthy donors also underwent measurement of serum creatinine at each time point, with results within the normal range. KTR are treated long-term with immunosuppressive drugs, including cyclosporine A, tacrolimus, and/or mycophenolate after receiving transplant (mean age at transplant: 32.4 years, range 13–58).

Both KTR and H donors were vaccinated with anti-SARS-CoV-2 mRNA-based vaccine (BNT162b2 Pfizer-BioNTech) and underwent a blood withdrawal at two time points: after the 2nd (T2, average 30.9 weeks, range 21–44) and 3rd dose (T3, average 10.6 weeks, range 2–20) (Figure 1B). One healthy control received the mRNA-based vaccine mRNA-1273 as a 3rd dose. The boost was injected soon after the T2 blood withdrawal (mean 4 days, range 0–10 before 3rd dose). Sera and peripheral blood mononuclear cells (PBMCs) were collected at T2 and T3 and cryo-preserved until the experiments were run up to measure: (1) anti-S total Ig, (2) anti-S nAbs, and (3) anti-SARS-CoV-2 specific CD4^+^ and CD8^+^ T cell responses. Specifically, blood samples for serum separation were collected in tubes without anticoagulant, while blood samples for PBMC isolation were collected in EDTA. PBMCs were obtained following the Ficoll HyPaque gradient centrifugation protocol. Moreover, at each time point, active SARS-CoV-2 infection was monitored by measuring total Ig anti-Nucleocapside antibodies (anti-N).

### 2.2. Total Anti-SARS-CoV-2 Ig Detection

Total anti-Spike SARS-CoV-2 Ig were measured by an Elecsys^®^Anti-SARS-CoV-2 Electrochemilumeniscence Immunoassay for Cobas analyzer (Roche Diagnostics, Basilea, Switzerland), based on a double-antigen sandwich assay format, according to the manufacturer’s instructions. In brief, 20 μL of serum was incubated with a mix of biotinylated and ruthenylated RBD antigen of S protein, and total anti-S SARS-CoV-2 Ig were reported in BAU/mL (WHO binding antibody units/mL). Total anti-Nucleocapside SARS-CoV-2 Ig were measured by an Elecsys^®^Anti-SARS-CoV-2 Electrochemilumeniscence Immunoassay for Cobas analyzer (Roche), based on the same double-antigen sandwich assay format of anti-S Ig, with the only difference that here a mix of biotinylated and ruthenylated recombinant N antigen was used. Concentrations ≥ 0.8 BAU/mL and a cut-off index > 0.99 were considered positive for anti-S and anti-N, respectively.

### 2.3. Neutralization Assay

SARS-CoV-2 S-pseudotyped lentiviral particles harboring the luciferase reporter gene were produced by transfection of HEK293T cells, aliquoted and stored at −80 °C until use. Pseudovirus titers were determined according to Neerukonda et al. [30].

A neutralization assay was performed according to Ni et al. and D’Apice et al. [31,32]. Briefly, 50 μL of SARS-CoV-2 S-pseudotyped Wuhan-Hu-1 isolate (*Wu*) or Omicron B.1.1.529 variant (*O*) lentiviruses (1 × 10^6^ Relative Light Units/mL, RLU/mL) were mixed with 50 μL of two-fold serially diluted heat-inactivated serum samples (ranging from 1/10 to 1/640) at 37 °C for 1 h. The mixtures (100 μL) were then transferred to 96-well plates seeded with HEK293-hACE2-hTMPRSS2 target cells (1.0 × 10^4^/well) stably expressing ACE2/TMPRSS2 proteins, 24 h before the pseudoviral infection. Then, 72 h later, cells were harvested and lysed using the Steady Glo luciferase assay system (Promega Corporation, Madison, WI, USA) to measure the luciferase activity by Victor multilabel plate reader. All samples were run in triplicates. The neutralization activity of each serum sample was calculated as follows: % Neutralization  =  (RLU_max_ − RLU_experimental_)/(RLU_max_ − RLU_min_) × 100, where RLU_max_ was the maximal infectivity calculated from untreated infected cells, RLU_experimental_ was calculated from infected cells treated with each serum dilution, RLU_min_ was the minimal infectivity calculated from uninfected cells. The nAb titers were expressed as the reciprocal of the highest serum dilution leading to 90% inhibition of RLUs (IC_90_). All samples with neutralization titers < 10 were considered negative and given an arbitrary value of IC_90_ = 5. The highest serum dilutions resulting in 90% reduction in luciferase production were referred to as pVNT_90_.

### 2.4. Functional T Cell Assays: Activation Induced Markers (AIM) and IFN-γ ELISA

SARS-CoV-2-specific T cell responses were measured by activation induced markers (AIM) assay [4,5,28] and IFN-γ ELISA after stimulation with peptide pools from *Wu* and Omicron *O* S proteins. More specifically, for the *Wu* strain we used a megapool (MP) containing 15-mer peptides (overlapping by 10 amino acids) and spanning the full-length of S protein. Overall, the *Wu* MP contained 253 peptides and was designed to activate either CD4^+^ and CD8^+^ T cells [4]. The *O* peptide pool was constituted by 83 overlapping 15-mer peptides, covering the S protein from B.1.1.529 Omicron variant (PepTivator SARS-CoV-2 Prot_S B.1.1.529 Mutation Pool, purchased from Miltenyi Biotech). As antigen/mitogen positive controls, a MP that includes both CD4^+^ and CD8^+^ T cell epitopes from cytomegalovirus (CMV), and phytohemagglutinin-L (PHA, Roche, Basilea) was used. MPs were diluted in DMSO at each peptide stock concentration of 1 mg/mL [4].

For AIM and ELISA assays, PBMCs from either H or KTR subjects were plated at 1–1.5 × 10^6^ cells/well in a U-bottom 96-well plate. Cells were then cultured for 20–24 h at 37 °C in presence of peptide pools tested at 1 µg/mL (final concentration of each peptide) or PHA at 2 µg/mL. As negative control, DMSO was added to the culture medium at the equivalent amount (final 0.2%) of that contained in MP experimental points.

After 24 h of incubation, before cell harvesting for FACS staining, 50 µL of culture supernatants were collected to measure the IFN-γ production by a standard ELISA sandwich assay, as previously described [33]. Briefly, IFN-γ was detected using purified and biotin-conjugated anti-IFN-γ Abs (purchased from Mabtech, Nacka Strand, Sweden). The sensitivity of the assay was 32 pg/mL. Briefly, cell supernatants were incubated for 2 h at room temperature on plates pre-coated with purified anti-IFN-γ Ab. An IFN-γ seven-point standard was included (62.5 to 4.000 pg/mL). Thereafter, plates were washed and the biotin-conjugated anti-IFN-γ Ab was added for 1 h at room temperature. Next, plates were washed and incubated with streptavidin peroxidase for 1 h at room temperature in the dark. Finally, an enzyme substrate solution was added and the OD_450_ measured. IFN-γ production was considered positive when production in stimulated condition (MP/PHA) was 2-fold greater than those of cells cultured in medium (DMSO) alone. Each condition was assayed in duplicates.

For FACS staining (AIM assay), cells were harvested, washed and resuspended in ice cold FACS buffer (PBS 1 × 0.5% BSA). CD4^+^ and CD8^+^ T lymphocyte activation was assessed by staining with a mixture of surface antibodies in the dark, at 4 °C for 30 min. After washing, the immunophenotyping was analyzed by FACSCanto II system and data were elaborated using FlowJO software (BD Biosciences). All experiments were performed in duplicates.

Live cells were gated based on propidium iodide (PI) exclusion. AIM positive cells were analyzed in the CD3^+^ gated cells, more specifically activated OX40^+^CD137^+^ and CD69^+^CD137^+^ cells were identified in the gate of CD4^+^ or CD8^+^ T cells, respectively. Percentage of AIM^+^ cells was calculated as net% of OX40^+^CD137^+^ (CD4^+^ T cells) and net% of CD69^+^CD137^+^ (CD8^+^ T cells) in response to spike/CMV MPs subtracting the % of AIM^+^ cells in response to DMSO. Details on fluorochrome-conjugated antibodies are reported in Appendix A.

### 2.5. Responsivity Criteria and Statistical Analysis

A two-tailed non-parametric Mann–Whitney test was used to compare the age between KTR and H donors, with no significant difference. T cell assays were considered positive when the AIM test and/or IFN-γ production in response to MP stimulation were at least two-fold greater than the baseline (DMSO control) values. Serum samples were considered unable to neutralize (neutralization titer < 10) when 90% inhibition of RLUs was not observed at the first dilution tested (1:10).

Statistical analyses were performed by comparing the percentage of AIM^+^ CD4^+^ or CD8^+^ T cells and IFN-γ production in response to MPs to those found at the baseline (DMSO) vaccination time points T2 and T3, and by comparing responses between T3 and T2 in the two groups of volunteers, by a two-tailed non-parametric paired Wilcoxon test. A *p* value < 0.05 was considered statistically significant. Similarly, statistical analysis for total anti-S Ig was performed by comparing concentrations measured at T3 and T2 time points in the two groups using a two-tailed non-parametric paired Wilcoxon test, with a significant *p* value < 0.05. For neutralization assay results, a two-tailed non-parametric Mann–Whitney test for unpaired observations and Wilcoxon test for paired observations were performed to compare pVNT_90_ between KTR and healthy individuals at T2 and T3. *p* values ≤ 0.05 were considered to be statistically significant. Correlations were assessed using the non-parametric Spearman rank correlation test. All statistical analyses were performed using GraphPad Prism 8.0.

## 3. Results

### 3.1. Total Antibody Response to SARS-CoV-2

It has been largely demonstrated that vaccine boosting induces a strong increase in humoral response against spike protein in the general population [4,34,35,36]. We found that the total anti-S Ig antibodies were significantly increased (*p* ≤ 0.05) at T3, after the 3rd dose of vaccine in both study cohorts (Figure 2A). In detail, in healthy subjects the mean value of antibody titer was 2508 BAU/mL at T2, after the 2nd dose, and 8593 BAU/mL at T3 (*p* < 0.05), while in KTR it was 126 BAU/mL at T2 and 4671 BAU/mL at T3 (*p* = 0.02) (Figure 2A). Notably, when we compared the total anti-S Ig levels between KTR patients and healthy subjects, significant differences were found at T3 after the booster dose, with the higher production detected in controls.

### 3.2. Neutralizing Antibody Response against SARS-CoV-2 Wuhan-Hu-1 and Omicron Strains

In line with other studies, we found that 100% of the healthy subjects produced neutralizing antibodies at T3 after vaccine boosting [20]. By contrast, in our cohort of KTR we registered an increase of seroconversion only in 57.14% (4/7) of recipients after the third vaccination, while in 42.86% (3/7) we were not able to detect any nAb activity against *Wu* strain (Figure 2B). Moreover, from the neutralization assay with *Wu* pseudotyped virus, we registered a 5.00-fold increase in neutralization titer after booster administration in KTR (GMT_T2_ = 5.95, GMT_T3_ = 29.72), and a 22.6-fold increase in healthy donors (GMT_T2_ = 11.89, GMT_T3_ = 269.1) (Figure 2B). When we used the Omicron strain pseudotyped virus to detect neutralizing activity, 42.86% (3/7) of KTR showed nAb in comparison to 75% (6/8) of H donors, and 2-fold increase of neutralization activity was detected in KTR cohort at T3 time point (GMT_T2_ = 5, GMT_T3_ = 10.00), while a fold increase of 4 has been registered in the healthy control group (GMT_T2_ = 5.95, GMT_T3_ = 23.78) (Figure 2C). Inter-individual changes in neutralization against *Wu* (Appendix A) or *O* variant (Appendix A) are reported before and after vaccine booster, showing an increase in neutralizing activity after 3rd vaccine dose either in H donors (*p* = 0.0156 *Wu*; *p* = 0.0312 *O*) or KTR cohort (*p* = 0.1250 *Wu*; *p* = 0.2500 *O*). To better compare the serum neutralization activity after the 3rd dose, the pVNT_90_ measured against *Wu* or *O* spike protein is shown in Appendix A. These results confirm that mRNA vaccine designed on the ancestral spike protein induce in these patients a humoral response directed mainly against the ancestral viral strain, while the emerging variant of concern (VOC) Omicron was only partially neutralized. A direct and significant correlation was observed between the magnitudes of total and neutralizing antibody responses in our study cohorts at T3 after the vaccination boost (*p* < 0.001, Figure 2D).

### 3.3. Analysis of Spike-Specific CD4^+^ and CD8^+^ T Cell Reactivity

Immune responsiveness to SARS-CoV-2 was assessed in vaccinated KTR (N = 7) and control subjects (N = 7) after stimulation of PBMCs with spike peptide pools containing *Wu* or *O* variant-derived sequences by the AIM assay with flow cytometry technology. The percentages of circulating T cells reacting to SARS-CoV-2 are measured within the subset of CD3^+^CD4^+^ cells expressing OX40 and CD137 surface markers and within the subset of CD3^+^CD8^+^ cells expressing CD69 and CD137 surface markers (Appendix A).

We observed in both KTR and healthy volunteers a statistically significant increase of percentage of *Wu*-specific CD4^+^ T cells after the 3rd dose of vaccine (T3), compared to the 2nd dose (T2) (*p* < 0.05) (Appendix A and Figure 3). In detail, in H subjects the median percentage of AIM^+^CD4^+^ T cells was 0.07% (range 0–0.32) at T2 and 0.32% (range 0.09–0.41) at T3. In KTR, median percentage was 0.08% (range 0.01–0.37) after 2nd dose and 0.33% (range 0.03–0.85) after the 3rd dose of vaccine, indicating a comparable response of these patients respect to healthy subjects. When we stimulated CD4^+^ T cells with pool of peptides from the Omicron spike, we observed lower percentages of reacting CD4^+^ T cells, with the increment from T2 to T3 being maintained in healthy controls (Figure 3A). Specifically, a median value of *O* specific CD4^+^ T cells in H subjects was 0.054% (range 0.003–0.056) at T2 and 0.10% (range 0–0.23) at T3, while in KTR remained unchanged at 0.12% (range 0–0.19) after the 2nd dose, and 0.08% (range 0.02–0.145) after the 3rd dose of vaccine. These findings demonstrate that, despite the immunosuppressive treatments, the adaptive CD4^+^ T cell response against Omicron was preserved in KTR, although it did not increase after the boost. The robustness of adaptive CD4^+^ T cell response was confirmed in response to CMV MP both in H (median 2.08, range 0.03–6.46) and KTR (median 0.45, range 0.1–2.25).

When we measured the adaptive immune response mediated by CD8^+^ T cells, we observed a low number of activated cells with no substantial changes in the density after the vaccination boost (Appendix A and Figure 3B). More specifically, the median percentage of T cells reacting against the *Wu* spike MP was 0.13% (range 0.01–0.4) at T2 and 0.1% (range 0–0.32) at T3 in H subjects. Similar percentage of AIM^+^ cells was observed in KTR. The CD69^+^CD137^+^CD8^+^ T cells were 0.1% (range 0–0.5) and 0.2% (range 0–0.32), respectively, at T2 and T3. Similar to CD4^+^ T cell response, we found a lower density of activated CD8^+^ T cells in response to *O* variant spike peptides in both groups, as AIM^+^ cells were 0.013% (range 0–0.17) at T2 and 0.005% (0–0.1) at T3 in healthy subjects, while they were 0% (range 0–0.01) at T2, and 0.07% (range 0–0.3) at T3 in KTR. By contrast, we detected a substantial number of CD8^+^ T cells reacting to CMV peptides in both patients and controls, thus excluding an immunosuppressive status of CD8^+^ T cells in KTR (H: median 2.26, range 0.5–7.96; KTR: median 4.1, range 0.35–7.5) (Appendix A and Figure 3B).

In conclusion, our data demonstrate that the SARS-CoV-2-specific CD4^+^ T cell response, in particular against the *Wu* strain, was significantly increased in both H and KTR subjects, after the 3rd dose of vaccine. Instead, the vaccine boost induced a low increment of CD4^+^ T cells against *O* variant in both groups. Lower CD8^+^ T cell reactivity was observed against both the *Wu* strain and *O* variant, after the 2nd and 3rd vaccinations, in both groups of analyzed subjects.

### 3.4. IFN-γ Production in Response to Spike Peptides and Correlation Analysis

To further assess the T cell activation elicited by MPs of *Wu* and *O* variant spike proteins, we measured the production of IFN-γ released in the cell supernatants using a classical ELISA sandwich assay (Figure 4A). The IFN-γ resulted slightly increased after specific stimulation with *Wu* MP in both KTR and H after the 2nd and 3rd vaccination. IFN-γ production in response to *O* peptide stimulation resulted much lower in both groups if compared with the cytokine level measured in response to MP from CMV. Interestingly, considering the overall cellular responses, we observed a profile of immune responsivity upon the vaccine boost ranging from 75 to 100% of H subjects and from 57 to 86% of KTR patients, thus confirming a reduced immune reactiveness of this group of transplanted patients upon vaccination (Figure 4B).

We further analyzed how the adaptive T cell response correlated with the titers of total and neutralizing anti-SARS-CoV-2 Wuhan-Hu-1 antibodies in our study cohorts. We found that after the 3rd dose of vaccine, 10/15 subjects, corresponding to 67% of volunteers (6 H, 4 KTR), showed an efficacious immune response, both in terms of IFN-γ and humoral (nAbs) response (Figure 4C,D). Interestingly, up to 80% (12/15, 8 H and 4 KTR) of vaccinated volunteers displayed a positive correlation between the T cell mediated and neutralizing antibody responses. Overall, these data suggest a clear correlation induced by the 3rd dose of vaccination between the T cell mediated cytokine release and antibody production in the immune protection against SARS-CoV-2.

## 4. Discussion

Since its beginning, the COVID-19 pandemic has set a severe threat to fragile patients due to the multiple comorbidities determining a poor outcome of the infection [9]. Immunocompromised individuals represent a population of particular interest, requiring an appropriate vaccination schedule as these subjects have weakened immune systems compared to healthy people. In fact, with regard to the vaccine campaign itself, it is important to pinpoint that the fragile populations were excluded from the initial trials [37]. Following the astonishing results of safety and efficacy in healthy populations, vaccine trials started to include immunocompromised individuals. Solid organ transplantation recipients (SOTRs) are at risk of severe COVID-19, with a higher rate of admission to intensive care units and higher mortality due to the immunosuppressant therapies [11]. Hence, we set a single center cohort study to investigate the immune response to BNT162b2 Pfizer-BioNTech mRNA vaccine over time in kidney transplanted patients. mRNA vaccine administration has been demonstrated to be partially effective in the induction of immune response in such patients, despite their long-term immunosuppressive anti-graft-rejection treatments [38], thus obtaining 80% reduction in the incidence of symptomatic COVID-19 versus unvaccinated [39], with booster doses having proven fundamental in guaranteeing the protection of this population of subjects [14,40,41]. Nevertheless, studies focused on the comprehensive immune response characterization following the 2nd and 3rd dose are always in need, in particular for the Omicron variant, since results reported up to now are controversial [20,42,43].

We selected a small KTR population aged between 20 and 73 years, in long-term treatment with immunosuppressive anti-rejection drugs used in clinical practice. We measured the humoral and cellular immune response at two time points: a long time after the 2nd dose of vaccine administration and a short time after vaccine boosting, in order to compare the booster vaccine administration-elicited immune response with a vaccinated matched control from healthy cohort.

We detected a significant increase in neutralization activity in KTR and healthy controls after vaccine boosting, using Wuhan-Hu-1 strain spike pseudotyped lentivirus, while we registered a strong impairment of the neutralization activity against the Omicron variant. These results indicated that the vaccine administering the ancestral SARS-CoV-2 S protein elicits an immune reaction but the aminoacidic mutations accumulated in the Omicron S protein favor immune evasion. Indeed, we detected a 22.6-fold increase in neutralizing activity in the healthy control group after the third dose administration, but it lowered to 4 when the Omicron S protein was used. A similar trend was seen in KTR, but at lower levels, probably because of their immunocompromised status. The described results are consistent with data reported by Benning et al. [20] and Yang et al. [27].

The cellular immune response against SARS-CoV-2 has already been deeply analyzed in previous studies [2,5,28,35,36]. Since the frequency of activated T cells in PBMC is low, we adopted the AIM assay, based on the upregulation of co-stimulatory molecules, that allows the detection of antigen-specific CD4^+^ and CD8^+^ T cells independently from the heterogeneity and amount of cytokine produced [4,5,28,35]. Moreover, the stimulation with large pools of overlapping peptides allows us the detect SARS-CoV-2-specific CD4^+^ and CD8^+^ T cell responses independently of HLA Class I and II haplotypes of analyzed subjects.

We found that the vaccine boosting produced a statistically significant increase in the frequency of *Wu*-specific CD4^+^ T cells, in both the analyzed cohorts, mirroring the humoral response. Omicron-specific CD4^+^ T cells were still detected, even if at a lower magnitude, in both groups, thus surprisingly also in KTR. Low but detectable CD8^+^ T cells were observed against *Wu* S antigen with frequencies similar to that found by others [23]. From the analysis of the IFN-γ, we detected an increased production in both groups at T3 against *Wu*, thus confirming the efficacy of vaccine boosting, however KTR showed levels of the cytokine higher than control subjects both at T2 and T3. As in AIMs assay, the magnitude of IFN-γ production detected against Omicron was lower in both groups. While the vast majority of the scientific studies aimed at measuring the cellular immune response elicited by vaccination were based on ELISA or ELISPOT instead that on activation markers, the results obtained were congruent with our observations [22,44].

Our results show that both the second and third dose of mRNA vaccine are able to elicit an adequate immune response to SARS-CoV-2, even if with differences in magnitude related to the viral strain and to the immunosuppressive therapy. In particular, the Omicron variant response is associated with lower antibody levels, lower CD4^+^ T expansion, and lower IFN-γ production. As expected, for both strains, the values measured in KTR were lower than in healthy subjects, but still present. Concerning the effect of booster vaccination on the kinetic of S-specific CD8^+^ T cells, a recent paper demonstrated that S-specific CD8^+^ T effector cell response, after the third dose, lasted 1–2 months and then underwent a contraction. Moreover, S-specific CD8^+^ T memory cells maintained long-term immunity but it was not very affected by third dose, remaining constant [45].

Our KTR patients are on long lasting treatments with corticosteroids, mycophenolate mofetil, cyclosporin A, tacrolimus, or sirolimus, that could explain the observed reduced immune responsiveness to vaccine administration. It is well documented that these drugs induce multiple immunosuppressive modifications [46].

The clinical follow-up did not show any difference in infection rates and outcomes, thus highlighting the importance of booster doses for the success of the vaccine campaign in the near future, particularly for immunocompromised patients. The absence of full-blown breakthrough infection (BI) in our small cohort does not let us neglect the evidence that a high rate of BI has been registered in SOTRs, after 2 or more doses of vaccine. Two recent studies reported 32% and 18.4% of BI in KTR [47] and SOTRs [48], respectively, nevertheless the number of BI may considerably depend on the time from transplantation and the therapies of patients. The risk of severe COVID-19 is influenced by several factors: (i) vaccine reduced efficacy, due to variants of interest (VOI) and VOCs less prone to be recognized by the vaccine instructed immune system [49]; (ii) waning immunity [50]. Hence, the detailed study of immune response to subsequent doses of vaccine may be useful to best address the choice of the competent authorities in making public health decisions.

This study has several limitations that may have influenced the observed results: a small sample size; variability of the time points evaluated within the cohort; an age difference between KTR and H donors; a wide age range in the KTR cohort; and the large time span of blood withdrawal. However, we are confident that the reported data might provide preliminary evidence on the vaccine-induced immunity in kidney transplanted patients, orienting prophylactic vaccine strategies against SARS-CoV-2 variants, and other emerging pathogens.

## Figures and Tables

**Figure 1 viruses-15-01132-f001:**
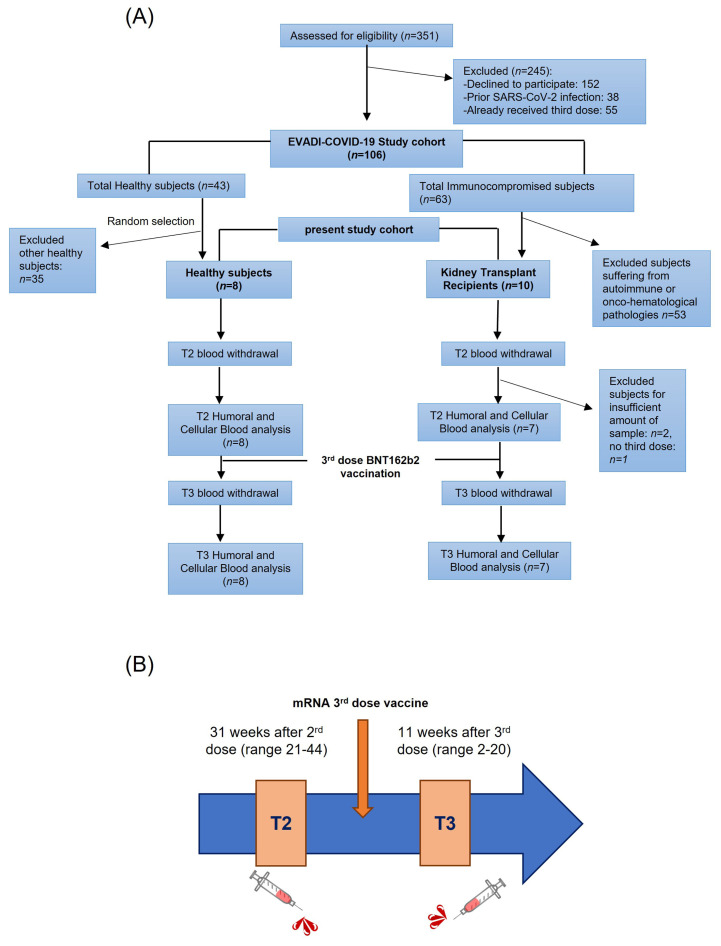
Study design. (**A**) Flow chart of the study population; *n*: number of samples. (**B**) Time points schematic representation.

**Figure 2 viruses-15-01132-f002:**
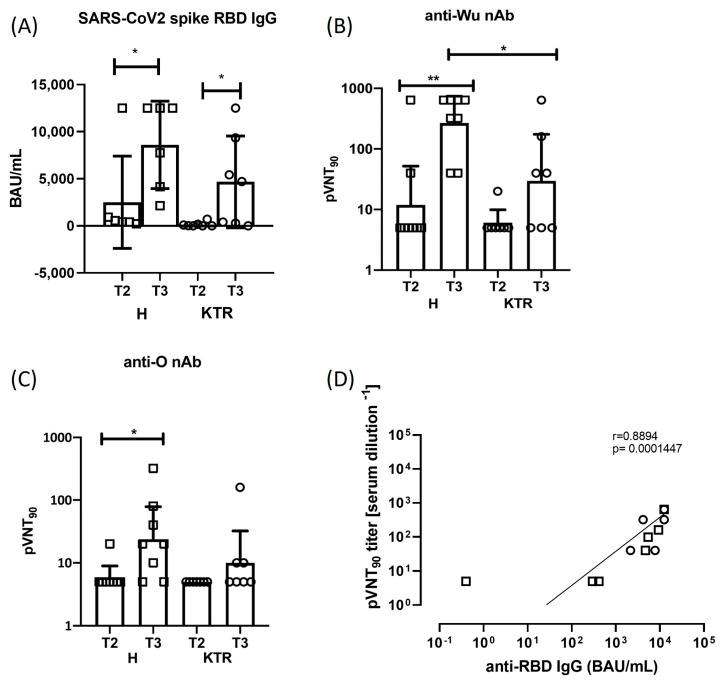
Spike antibody response. Measurement of total Ig and neutralizing anti-spike antibody titers in healthy subjects (H) and kidney transplanted patients (KTR) after the 2nd and 3rd vaccine dose (T2 and T3 time points). Statistical analysis was performed by paired Wilcoxon test (* *p* ≤ 0.05). (**A**) Total anti-Spike/RBD Ig quantitative detection in sera collected at T2 and T3 time points, measured by Elecsys^®^ Anti-SARS-CoV-2 S assay. BAU/mL: Binding Antibody Units. (**B**,**C**) Neutralization of *Wu* and *O* variant spike-expressing pseudotyped lentiviral particles by antibodies in sera of KTR and H donors at T2 and T3 time points. The graphs show the geometric means (GMT) +/− 95% confidence intervals. (**D**) Correlation between the anti-*Wu* spike neutralizing and total antibodies in H (squares, N = 6) and KTR (circles, N = 7) vaccinated volunteers at T3 is shown. Correlation was assessed using the non-parametric Spearman rank correlation coefficient. Mann–Whitney test was used to compare data in panels (**B**,**C**), with a *p* value < 0.05 considered statistically significant (* *p* < 0.05; ** *p* < 0.005). All statistical analyses were performed with GraphPad Prism 8.0.

**Figure 3 viruses-15-01132-f003:**
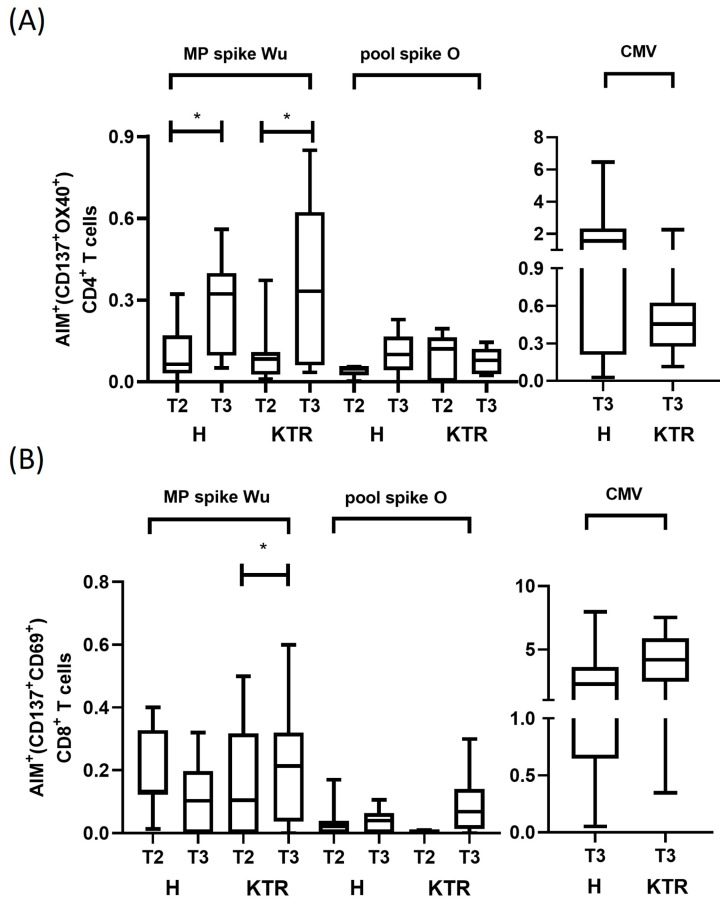
Wuhan-Hu-1 and Omicron spike-specific CD4^+^ and CD8^+^ T cell response by AIM assay. (**A**) Spike-specific CD4^+^ T cells were evaluated in KTR and healthy controls by AIM assay after the 2nd and 3rd vaccine dose (T2 and T3 time points). The percentage of OX40^+^CD137^+^CD4^+^ T cells were measured in PBMCs after stimulation with spike protein overlapping peptides derived from ancestral Wuhan-Hu-1 strain (*Wu*) and Omicron variant of concern (*O*) by flow cytometry. Percentage of OX40^+^CD137^+^CD4^+^ T cells detected in healthy subjects (H, N = 7) and kidney transplanted patients (KTR, N = 7). Data are plotted as NET value, subtracted of DMSO background values (mean value 0.06%, range 0–0.16). (**B**). Spike-specific CD8^+^ T cells were evaluated in KTR and healthy controls by AIM assay after the 2nd and 3rd vaccine dose (T2 and T3 time points). The percentage of CD69^+^CD137^+^CD8^+^ T cells were measured in PBMCs after stimulation with spike protein overlapping peptides derived from ancestral Wuhan-Hu-1 strain (*Wu*) and Omicron variant of concern (*O*) by flow cytometry. Percentages of CD69^+^CD137^+^CD8^+^ T cells detected in healthy subjects (H, N = 7) and kidney transplanted patients (KTR, N = 7). Data are plotted as NET value, subtracted of DMSO background values (mean value 0.23%, range 0.027–1.26). Paired Wilcoxon test was used to assess statistical significance, with a *p* value < 0.05 considered significant (* *p* < 0.05). All statistical analyses were performed using GraphPad Prism 8.0.

**Figure 4 viruses-15-01132-f004:**
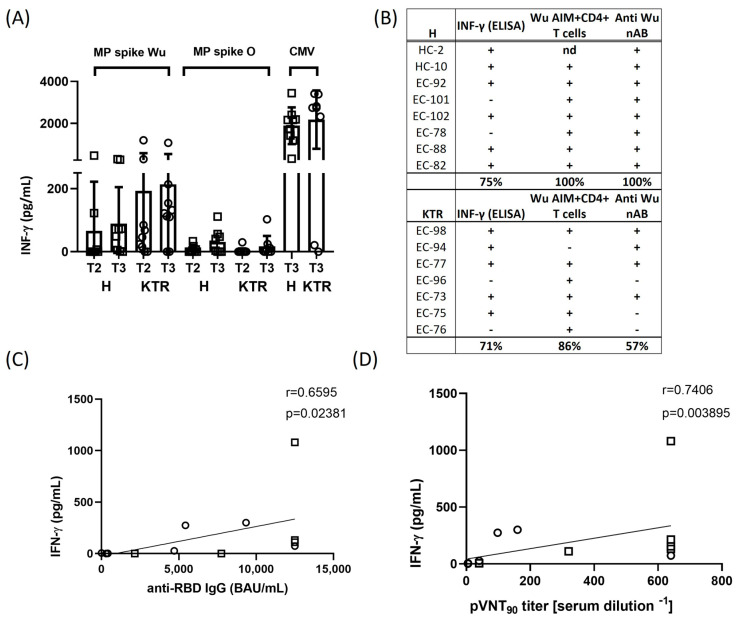
IFN-γ cytokine production and correlation between adaptive cellular and humoral responses. Spike protein-induced IFN-γ production was assessed after 2nd and 3rd vaccine dose (at T2 and T3 time points) in healthy volunteers (H) and kidney transplanted patients (KTR). (**A**) IFN-γ was detected in the cell supernatants by enzyme-linked immunosorbent assay (ELISA) after 24 h stimulation of PBMCs with overlapping peptides derived from ancestral Wuhan-Hu-1 strain (*Wu*) and Omicron variant of concern (*O*). IFN-γ data were subtracted of background DMSO values. Each bar indicates the mean of value and standard deviation. (**B**) Profile of immune responsiveness to ancestral Wuhan-Hu-1 strain after the vaccine booster dose (T3). Immune response against SARS-CoV-2 ancestral Wuhan-Hu-1 strain, assessed as either an AIM test and/or IFN-γ production, was considered positive when it was at least two-fold greater than the baseline (DMSO control) values. The cut-off for positive nAb response was a neutralization titer ≥ 10. Correlation between IFN-γ production and total anti-RBD antibody titers (**C**) and between IFN-γ production and anti-Wuhan-Hu-1 nAb antibody titers (**D**) in 5 H (squares) and 7 KTR (circles) and 7 H (squares) and 7 KTR (circles) subjects, respectively. Correlations were performed using the non-parametric Spearman rank correlation coefficient to compare the magnitudes of total and neutralizing antibodies against Wuhan-Hu-1 strain after the 3rd vaccination. All statistical analyses were performed by GraphPad Prism 8.0, and a *p* value < 0.05 was considered statistically significant.

**Table 1 viruses-15-01132-t001:** Study cohort enrolled in EVADI-COVID-19 project.

Code	Age (ys)/ Sex	Volunteer Condition	Age at Transplant	Years after Transplant	Underlying Disease	T2 Serum Creatinine (mg/dL)	T2 Creatinine Clearance (mL/min)	T2 CKD Stage	T3 Serum Creatinine (mg/dL)	T3 Creatinine Clearance (mL/min)	T3 CKD Stage	T2 (Weeks from 2nd Dose)	T3 (Weeks from 3rd Dose)	Weeks between T2 and T3	Immunosuppressive Treatment
EC-73	44 ys/F	KTR	28	16	arterial hypertension	0.87	84	2-T	0.86	85	2-T	22	17	2.4	TAC 8mg/day
EC-75	73 ys/M	KTR	58	15	arterial hypertension	1.39	50.5	3a-T	1.34	50.6	3a-T	21	20	2.8	TAC 4 mg/day; MMF 500 mg/day
EC-76	30 ys/M	KTR	24	6	IgA nephritis	1.54	62	2-T	1.76	66.5	2-T	26	15	2.1	CsA 60 mg/day; MMF 900 mg/day; PDN 5 mg/day
EC-77	20 ys/F	KTR	13	7	nephritis	0.68	127	1-T	0.83	103	1-T	22	13	1.8	TAC 4 mg/day; MMF 500 mg/day
EC-94	54 ys/M	KTR	36	18	obstructive renal failure	2.22	34	3b-T	2.32	33	3b-T	27	7	1.0	SIR 1 mg/day; mPRED 4 mg/day
EC-96	30 ys/F	KTR	23	7	ADPKD	1.66	42	3b-T	1.65	41	3b-T	29	13	1.8	CsA 170 mg/day; MMF 720 mg/day; mPRED 4 mg/day
EC-98	57 ys/F	KTR	45	12	arterial hypertension	1.2	53	3a-T	1.06	61	2-T	31	5	0.7	TAC 1.5 mg/day; MMF 1000 mg/day; PDN 5 mg/day
EC-78	68 ys/M	HC	-	-	none	-	-	-	-	-	-	37	17	2.4	none
EC-82	63 ys/M	HC	-	-	none	-	-	-	-	-	-	32	8	1.1	none
EC-88	43 ys/M	HC	-	-	none	-	-	-	-	-	-	39	17	2.4	none
EC-92	37 ys/M	HC	-	-	none	-	-	-	-	-	-	41	10	1.4	none
EC-101	53 ys/F	HC	-	-	none	-	-	-	-	-	-	44	6	0.9	none
EC-102	33 ys/F	HC	-	-	none	-	-	-	-	-	-	43	6	0.8	none
HC-2	41 ys/F	HC	-	-	none	-	-	-	-	-	-	24	4	0.6	none
HC-10	49 ys/F	HC	-	-	none	-	-	-	-	-	-	25	2	0.3	none

ys: years; transplant.: transplantation; CKD: chronic kidney disease; KTR: kidney transplant recipients; HC: healthy control; ADPKD: Autosomic Dominant Polycystic Kidney Disease; MMF: mycophenolate mofetil; TAC: tacrolimus; PDN: prednisone; mPRED: methylprednisolone; SIR: sirolimus; CsA: cyclosporine A.

## Data Availability

Not applicable.

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
