# Peer review of "Booster Dose of SARS-CoV-2 mRNA Vaccine in Kidney Transplanted Patients Induces Wuhan-Hu-1 Specific Neutralizing Antibodies and T Cell Activation but Lower Response against Omicron Variant"

_viruses, 2023, doi:10.3390/v15051132_

Round 1
Reviewer 1 Report
In this paper, the authors have analysed both humral and cellular response in response to COVIID-19 mRNA vaccines. They compared the response againts the original S protein of SARS-CoV-2 and againts Omicron variant.
Major comments: the major limitation of the study is the very low number of patients enrolled. The number of patients and controls should be indicated in the abstract. Additionnaly, among the 8 patients with kidney transplanted recipients (KTR), only 7 received the third dose, therefore I think that the last one can not be included in the study. Similarly page 3, in the figure 1, dots are indicated for 7 patients at T3, but in the text, line 272, percentages are indicated for 8 patients at T3.
- More details concerning statistical analysis should be indicated, The authors indicate that they use unpaired Student-t test for comparison responses between T2 and T3, certainly because they have no level for the 8 patients. However, it is not indicated whether the distribution is normal for this very small series of patients, allowing the use of the t-test. But in my opnion, it would be more rigourous to use paired statistical tests....In the same way, in supplementary fig S2, it is indicated that the comparison between T2 and T3 included 8 patients, but levels are indicated for only 3 patients for controls and 5 for KTR. So, I suggest, for each experiment, that authors indicated the number of patients tested.
- in the paragraph analysing the production of IFN gamma, (page 6, lines 374 - 382), authors mixed Controls and TKR. This is an uncommon approach.
- more details concerning the control group should be added and statistical comparison with the patients shoul be provided. For example, it is only indicated at te end of the discussion that age were significantly different between controls ad patients.
Minor comments: Table 1 should be simplified; For example, it is indicated no for each patients in the column diabetes; It should be easier to indicate in the general description of the patients that none were diabetic. Is it useful to indicate both serum creatinine levels and clearnace ? In contrast it should be useful to indicate creatinine clearence of the controls.
- Why "PDN 5 mg/day" in the top of the values concerning the controls, in the table 1 ?
- page 2, line 250: for antibody titer in BAU, is it essential to put 2 digits after the point, if we consider the reproducibility and the sensitivity of the method?
- the flow chart of the study (Fig S1) seems important and should not be considered only as a supplementary data.
- Why to present only the correlations between tha anti-Wu spike neutralysing antibody in fig 1D for controls, ? It should be intersting to add the levels of the patients.
Author Response
Comments and Suggestions for Authors
In this paper, the authors have analysed both humral and cellular response in response to COVIID-19 mRNA vaccines. They compared the response againts the original S protein of SARS-CoV-2 and againts Omicron variant.
Major comments: the major limitation of the study is the very low number of patients enrolled. The number of patients and controls should be indicated in the abstract. Additionnaly, among the 8 patients with kidney transplanted recipients (KTR), only 7 received the third dose, therefore I think that the last one can not be included in the study. Similarly page 3, in the figure 1, dots are indicated for 7 patients at T3, but in the text, line 272, percentages are indicated for 8 patients at T3.
Author’s response: We thank the Reviewer for the helpful comments. As reported in the actual revised Figure 1 (Study design), this study was designed on 351 eligible patients. The final number of enrolled subjects (7 KTR and 8 H) is chiefly due to the unwillingness to be enrolled (n=152) during the pandemic, since movement restriction and fear of infection influenced the patient decision and, partially, to prior infection (n=38). Among the 63 willing immunocompromised patients, we excluded 53 immunocompromised patients for the non-homogeneity of their pathologies, as they were patients with onco-hematological or autoimmune diseases under different pharmaceutical therapies. To have a homogeneous patient cohort, we selected 10 KTR and then excluded: n=2 for insufficient amount of sample. In the revised version of the manuscript, we have also excluded n=1 KTR since no third dose was administered, and removed all the relative data as suggested by the Reviewer, reaching the final number of 7. According to Reviewer #2’s suggestion, we have re-submitted our study as brief report, clearly indicating the number of subjects enrolled in the abstract (line 23). As brief report, we have overall re-organized all the figures. Concerning percentages, we apologize for the mistake and accordingly we have indicated the correct values (lines 286-299) and also revised the actual Figure S1 (Individual nAb response).
- More details concerning statistical analysis should be indicated, The authors indicate that they use unpaired Student-t test for comparison responses between T2 and T3, certainly because they have no level for the 8 patients. However, it is not indicated whether the distribution is normal for this very small series of patients, allowing the use of the t-test. But in my opnion, it would be more rigourous to use paired statistical tests....In the same way, in supplementary fig S2, it is indicated that the comparison between T2 and T3 included 8 patients, but levels are indicated for only 3 patients for controls and 5 for KTR. So, I suggest, for each experiment, that authors indicated the number of patients tested.
Author’s response: We thank the Reviewer for the suggestion and accordingly we have re-analysed data graphed in the current revised Figures 2A, 3A-B, and 4A using paired statistical tests and provided more details concerning statistical analysis at lines 249 and 252. Concerning supplementary Fig S2, now revised Figure S1 (Individual nAb response), from the Reviewer’s comment it emerges that the graphical representation is misleading. The total number of analysed samples is indicated below each graph (n=8 H, n=7 KTR). As some samples have the same pVNT90 value, they graphically overlap, giving the impression of a reduced number of values.
- in the paragraph analysing the production of IFN gamma, (page 6, lines 374 - 382), authors mixed Controls and TKR. This is an uncommon approach.
Author’s response: We thank the Reviewer for the observation. However, correlation analyses between IFN-gamma production and total anti-RBD antibody titers (Figure 4C) and between IFN-gamma production and anti-Wuhan-Hu-1 nAb antibody titers (Figure 4D) were performed in order to investigate the overall efficacy of a mRNA-based vaccine boosting at inducing immunity in a general population, apart from long lasting treatments with drugs and the immune profile of vaccinated individuals. However, the graphical representation of H with squares and KTR with circles symbols should be helpful at discriminating the two groups.
- more details concerning the control group should be added and statistical comparison with the patients shoul be provided. For example, it is only indicated at te end of the discussion that age were significantly different between controls ad patients.
Author’s response: We thank the Reviewer for the suggestion and accordingly we have provided the results from the statistical comparison between the control group and KTR, such as mean age ± SD, median, IQR, number of male and female, in “Cohort Description and Sample Collection” paragraph (lines 116-119), clarifying the statistical test used for the analysis in “Responsivity Criteria and Statistical Analysis” paragraph (lines 240-241). Moreover, performing the two-tailed non-parametric Mann–Whitney test, the age difference between KTR and H donors was not significant.
Minor comments: Table 1 should be simplified; For example, it is indicated no for each patients in the column diabetes; It should be easier to indicate in the general description of the patients that none were diabetic. Is it useful to indicate both serum creatinine levels and clearnace ? In contrast it should be useful to indicate creatinine clearence of the controls.
Author’s response: As suggested by the Reviewer, we have simplified Table 1 removing the “diabetes” column, keeping the statement “No patient was affected by diabetes” at lines 129-130. Both values of creatinine levels and clearance were provided in order to monitor post-transplant kidney function as reported at lines 131-132. As stated at lines 135-137, healthy donors also underwent measurement of serum creatinine at each time point, with results in normal range (data not shown). We have also removed from the table data concerning one KTR as this patient was excluded from the analysis as suggested by the Reviewer.
- Why "PDN 5 mg/day" in the top of the values concerning the controls, in the table 1 ?
Author’s response: The “PDN 5mg/day” value showed at the top refers to EC-98 KTR, as clearly presented now in the revised version of Table 1.
- page 2, line 250: for antibody titer in BAU, is it essential to put 2 digits after the point, if we consider the reproducibility and the sensitivity of the method?
Author’s response: We thank the Reviewer for the suggestion. Accordingly, we have removed the 2 digits after the point in the antibody titers at lines 265-266.
- the flow chart of the study (Fig S1) seems important and should not be considered only as a supplementary data.
Author’s response: We thank the Reviewer for this suggestion. Accordingly, we have moved the flow chart of the study into the text of the manuscript and renamed it as Figure 1 (lines 95 and line 143).
- Why to present only the correlations between tha anti-Wu spike neutralysing antibody in fig 1D for controls, ? It should be intersting to add the levels of the patients.
Author’s response: Figure 1D, now revised Figure 2D, shows the correlation between the anti-Wu spike neutralizing and total antibodies in all vaccinated volunteers at T3, with squares indicating H and circles indicating KTR.
Reviewer 2 Report
The paper by Del MAstro and colleagues is a cohort study about the response to SARS-CoV-2 vaccine in kidney transplant recipients
The introduction is informative and sufficently sinthetic and the aims of the study are clearly explained
The methods section is suffiently clear, but can be improved (see minor points)
The results section appears as well presented, sinthetic and clear, and data are of interest, despite being objectively influenced by the extremely low number of subjects
The discussion is probably too long and should be significantly made more synthetic.
In general terms, the paper is a sufficiently good quality, and the techniques which have been used are adequate. But the number of subjects is really too small, and this
overshadows also the other limitations, related to the heterogenicity of the cohort. It is also hard to understand how the HC group, which was selected from a large cohort, is so
different from the KTR group.
These data could be of interest if enriched with more subjects. In this terms, this paper could be re-formatted for a short communication, but in my opinion enlarging the study cohort
is a better option. I will suggest a major revision to allow the authors try this option.
Some minor points:
- line 157: please add complete name for both assays, and possibly some references to manufacturers protocols.
Please note that both methods recognize total antibodies, not IgGs.Also, please verify if cutoff for the N-assay is 1, not 0.8
- line 193: AIM assay is a quite recent technique. Adding some more general explanation and a reference would help the readers
- line 210: the ELISA is described in a too short and non-organized way. Please expand and structure the period in a better form.
- line 230: how were these cutoffs identified? please clarify
- line 261: is it ELISA or Elecsys ?
- line 537: as these are fundings, they should be added in the appropriate section
- line 564: please correct first author name's spelling
- line 592: please correct sixth author name's spelling
- line 632: the authors are missing
Author Response
Comments and Suggestions for Authors
The paper by Del MAstro and colleagues is a cohort study about the response to SARS-CoV-2 vaccine in kidney transplant recipients
The introduction is informative and sufficently sinthetic and the aims of the study are clearly explained
The methods section is suffiently clear, but can be improved (see minor points)
The results section appears as well presented, sinthetic and clear, and data are of interest, despite being objectively influenced by the extremely low number of subjects
The discussion is probably too long and should be significantly made more synthetic.
In general terms, the paper is a sufficiently good quality, and the techniques which have been used are adequate. But the number of subjects is really too small, and this
overshadows also the other limitations, related to the heterogenicity of the cohort. It is also hard to understand how the HC group, which was selected from a large cohort, is so
different from the KTR group.
These data could be of interest if enriched with more subjects. In this terms, this paper could be re-formatted for a short communication, but in my opinion enlarging the study cohort
is a better option. I will suggest a major revision to allow the authors try this option.
Author’s response: We are grateful to the Reviewer for the overall positive evaluation of our study. According to Reviewer’s suggestion, we have re-submitted our study as brief report, re-organizing all the figures and shortened the Discussion section. We are aware of the small number of enrolled patients. However, we believe that the reported data could be of interest in the field, providing evidences on the vaccine-induced immunity in KTR patients. As reported in the actual revised Figure 1 (Study design), this study was designed on 351 eligible patients. The final number of enrolled subjects (7 KTR and 8 H) is chiefly due to the unwillingness to be enrolled (n=152) during the pandemic, since movement restriction and fear of infection influenced the patient decision and, partially, to prior infection (n=38). Among the 63 willing immunocompromised patients, we excluded 53 immunocompromised patients for the non-homogeneity of their pathologies, as they were patients with onco-hematological or autoimmune diseases under different pharmaceutical therapies. To have a homogeneous patient cohort, we selected 10 KTR and then excluded: n=2 for insufficient amount of sample. In the revised version of the manuscript, we have also excluded n=1 KTR since no third dose was administered and removed all the relative data, reaching the final number of 7.
Concerning differences between groups, we performed the two-tailed non-parametric Mann–Whitney test to analyse the age difference between KTR and H donors, with not significant difference observed. We provided values of mean age ± SD, median, IQR, number of male and female, in “Cohort Description and Sample Collection” paragraph (lines 116-119), clarifying the statistical test used for the analysis in “Responsivity Criteria and Statistical Analysis” paragraph (lines 240-241).
Some minor points:
- line 157: please add complete name for both assays, and possibly some references to manufacturers protocols.
Please note that both methods recognize total antibodies, not IgGs.Also, please verify if cutoff for the N-assay is 1, not 0.8
Author’s response: As suggested, we have added the complete name for both assays, indicating the cutoff for the N-assay of 0.99 (lines 169-170), overall amended the text changing IgG into Ig.
- line 193: AIM assay is a quite recent technique. Adding some more general explanation and a reference would help the readers
Author’s response: According to the Reviewer’s suggestion, we have clarified in “Functional T Cell Assays: Activation Induced Markers (AIM) and IFN-g ELISA” paragraph the AIM assay, citing Tarke et al Cell 2022, Mateus et al 2021, and Grifoni et al Cell 2020 at line 197.
- line 210: the ELISA is described in a too short and non-organized way. Please expand and structure the period in a better form.
Author’s response: As suggested by the Reviewer, we have provided additional details of the ELISA protocol (lines 216-224).
- line 230: how were these cutoffs identified? please clarify
Author’s response: We have clarified at line 243-245 as fellows: “Serum samples were considered not able to neutralize (neutralization titre <10) when 90% inhibition of RLUs was not observed at the first dilution tested (1:10)”.
- line 261: is it ELISA or Elecsys ?
Author’s response: We apologize for this mistake and corrected as follows: “measured by Elecsys® Anti-SARS-CoV-2 S assay” (line 275).
- line 537: as these are fundings, they should be added in the appropriate section
Author’s response: According to the Reviewer’s observation, we moved the following statement to line 519-522: “This project has been funded in part with Federal funds from the National Institute of Allergy and Infectious Diseases, National Institutes of Health, Department of Health and Human Services, under Contract 75N93019C00065 to A.S.”.
- line 564: please correct first author name's spelling
Author’s response: The first author’s name of reference #6 is: (Corine H.) GeurtsvanKessel as correctly reported at line 564.
- line 592: please correct sixth author name's spelling
Author’s response: We thank the Reviewer for the observation and have accordingly corrected the sixth author name’s spelling.
- line 632: the authors are missing
Author’s response: We thank the Reviewer for the observation and have accordingly edited the relative reference.
Round 2
Reviewer 1 Report
no more comment
Reviewer 2 Report
The paper has been improved in some parts and in the current form can be accepted as a brief report as indicated